# Analyzing Passenger Flows in an Airport Terminal: A Discrete Simulation Model

Cristina Oprea *, Mircea Rosca, Eugen Rosca, Ilona Costea, Anamaria Ilie, Oana Dinu and Aura Ruscă

Transport Faculty, National University for Science and Technology Politehnica, Bucharest, Spl. Independentei, No 313, RO-060042 Bucharest, Romania; mircea.rosca@upb.ro (M.R.); eugen.rosca@gmail.com (E.R.); ilona.costea@upb.ro (I.C.); anamaria.ilie@upb.ro (A.I.); oana.dinu@upb.ro (O.D.); aura.rusca@upb.ro (A.R.)
* Correspondence: cristina.oprea@upb.ro; Tel.: +40-744774354

**Abstract:** This paper introduces a simulation model designed as a decision-making tool to assess and analyze various crowd management strategies with a focus on enhancing sustainability in airport operations. This model specifically addresses the challenges and risks associated with managing passenger flows within airport terminals. By simulating different scenarios, the model aims to provide valuable insights into how to effectively handle crowd dynamics and enhance overall terminal efficiency, safety, and sustainability. This case study was conducted at Henri Coanda International Airport, ARENA 12 simulation software being used in order to model the passenger flows within the airport terminal. Two scenarios were considered: The first one involves maintaining a fixed number of security and check-in desks for the two airline groups. In contrast, the second scenario allows for a variable number of security and check-in desks for the same airline groups. By optimizing resource allocation and minimizing waiting time, this model contributes to more sustainable airport management operations. Three measures of performance (MOPs) were selected to assess the system activity: the average passenger waiting time, the average passenger number queue length, and the average utilization rate. Comparing the results, we concluded that the second scenario shows a relative improvement in almost all performance measures when compared to the first scenario.

**Keywords:** airport; terminal; passenger flow; simulation; public transport

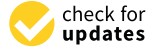



## 1. Introduction

The aim of this paper is to develop and apply a simulation model for assessing and optimizing crowd management strategies in airport terminals, with a focus on enhancing operational efficiency, safety, and sustainability under variable passenger flow conditions.

Airport terminals must be designed to handle the increasing volume of passengers while ensuring an adequate level of service. This involves managing waiting times in queues, reducing overcrowding, and minimizing delays. As air travel continues to grow, airports face the challenge of scaling up their operations to meet rising passenger numbers without compromising service quality. This requires a delicate balance between efficiency and comfort, where passengers experience smooth and timely movement through the airport from check-in to boarding [1].

As passenger numbers increase, airports face significant logistical challenges in managing the flow of passengers through terminals. Efficient passenger movement is critical to reducing congestion and minimizing waiting times. Airports must consider the layout of the terminal, the placement of security checkpoints, and the integration of technology to optimize passenger flow. Solutions such as automated check-in kiosks, real-time queue management systems, and dynamic signaling can help direct passengers more efficiently. Furthermore, airports may need to explore the expansion of terminal spaces or the construction of new facilities to accommodate the growing number of passengers.

Due to congestion in the check-in offices and security check area, safety threats have become prevalent in airport terminals. Consequently, effective crowd management strategies are essential to organize passenger movement during operating times, minimizing waiting times in lines.

The goal for airports is to strike a balance between operational efficiency and passenger experience. While it is essential to process passengers quickly and efficiently, it is equally important to ensure that their journey through the airport is pleasant and stress-free. This involves providing adequate seating, entertainment, and dining options, as well as ensuring that the terminal environment is clean, comfortable, and welcoming. Airports that effectively maintain this balance will not only accommodate the growing demand of passengers, but also strengthen their reputation and competitive edge in the global market.

## 2. Literature Background

The study of passenger flows within airport terminals has attracted considerable attention in recent years, with various researchers examining terminal operations from a range of perspectives, including transit capacity, terminal design, passenger flow estimation, and safety and security concerns. Studies have been conducted to explore the ways in which airports can optimize operations, improve the passengers' experience, and enhance overall efficiency.

Relógio and Tavares (2023) [2] analyzed the degree of client satisfaction among airline passengers by three factors: the waiting time and service at the airline office, the comfort during the trip, and the empathy of the cabin staff. The theoretical and practical implications of this study highlight the necessity of emphasizing service quality, aiming for continuous improvement across the air transport sector.

Koh, Cal, and Diaz (2011) [3] conducted an in-depth analysis of amenities preferred by passengers in public transport terminals. They identified patterns in passenger preferences regarding terminal conditions and amenities, which have implications for improving terminal planning and design. Their study emphasizes that passenger satisfaction is closely linked to the quality and availability of services provided in transit terminals, suggesting that such considerations should be central in terminal development.

Nommika and Antov (2017) [4] focused on developing capacity assessment models specifically for airport terminals. They introduced an innovative approach for estimating terminal capacity by analyzing the dynamic patterns of passenger flows. Their research offers key insights into optimizing capacity to handle fluctuating passenger volumes, which is crucial for effective airport terminal management.

The integration of multiple transport modes within passenger terminals has also been a subject of exploration. Margarita et al. (2016) [5] provided an extensive review of eight intermodal public transport terminals, focusing on design features that facilitate the seamless integration of different modes of transportation. Their work highlighted the importance of innovative designs in enhancing the efficiency and functionality of these terminals.

Estimating and managing passenger flow within terminals has also been the focus of several studies, such as those by Ahn et al. (2017), Liu and Chen (2017), and Rusca et al. (2013) [6–8]. These studies used a variety of methodologies to analyze passenger flows. Ahn et al. applied a classical four-step model to identify critical congestion points within terminals. Liu and Chen used an evolutionary algorithm based on neural networks to predict passenger flows, while Rusca et al. explored the relationship between train schedules and ticket office workloads in railway terminals.

Kirlangicoglu (2015) [9] advanced the study of terminal design by investigating how fundamental passenger needs—such as accessibility, safety, comfort, and satisfaction—influence the design and operation of terminals. This work highlights the importance of aligning terminal design with passenger expectations to improve the overall travel experience.

Discrete Event Simulation (DES) has been adopted as a tool for studying passenger flow management within airport terminals. Alodhaibi et al. (2017) [10] used DES to analyze processes from curbside to boarding, employing ExtendSim V9.2 software to identify

bottlenecks and optimize passenger flow. Their findings underline the importance of simulations in enhancing airport planning and improving operational efficiency. Similarly, Guizzi et al. (2009) [11] applied Discrete Event Theory to develop a framework for predicting delays and optimizing the management of check-in and security processes. Their work emphasizes the significance of simulation models in minimizing delays and improving the passenger experience.

In a more recent study, Anagnostopoulou et al. (2024) developed a decision-making tool that incorporates various strategies to manage passenger flows in airport terminals. Their research demonstrates that optimizing passenger routing through modern technological solutions is the most effective method for managing crowds in airports with a capacity of 800 passengers per hour. The model they introduced is adaptable and can be adjusted to meet the changing needs of any airport terminal [12].

Overall, these studies demonstrate the value of various methodologies, from capacity models to simulation tools, in enhancing the efficiency and passenger experience within airport terminals. Discrete simulation models, in particular, play a crucial role in identifying bottlenecks and streamlining passenger flow, making them essential for improving terminal operations and planning.

## 3. Modeling Passenger Flows

A public transport terminal can be categorized into three main functional areas, each corresponding to different aspects of passenger activities:

- Access Interface: This area encompasses the initial point of interaction where passengers arrive at the terminal and prepare to use their chosen mode of transport. Key activities in this zone include vehicle movement and parking, as well as designated spaces for buses and taxis.
- Processing Area: Located within the terminal, this area is where passengers—whether arriving, departing, or in transit—are handled and serviced. For high-capacity terminals such as airports, it is beneficial to separate the flows of arriving and departing passengers to enhance efficiency. The main activities in this area include the following: passenger information—providing essential updates and guidance to passengers; ticketing services; retail and services—offering shopping and dining options; waiting—designated spaces for passengers to wait before boarding; and check-in and security control.
- Transport Interface: This final component involves the transition from the terminal to the transport modes. It includes the following: boarding platforms—areas where passengers board their chosen transport modes, whether they are starting their journey from the terminal or continuing from transit; alighting areas—spaces where passengers disembark from their transport and exit the terminal if their destination is the terminal.

Each of these three areas plays a critical role in ensuring a seamless and efficient experience for passengers as they move through the terminal.

## 4. Case Study

### 4.1. Bucharest Henri Coanda International Airport

The present research aims to develop a simulation model able to optimize, in a logical and rational way, the check-in desks for domestic flights, the security control for all flights, and the boarding for domestic flights inside the departure terminal at Bucharest Henri Coanda International Airport. The arrival of passengers in the terminal is a discreet process. The modeling of the passenger flow in the terminal is suitable for the use of a discrete-type process model. The software package chosen by the authors is Rockwell ARENA Simulation, which allows for the creation of a modular logical model that is easy to represent [11,13,14].

Several studies in the literature have utilized the Rockwell ARENA Simulation tool, having significant success in modeling and capturing the dynamics of real-life experimental data. For example, Guizzi et al. [11] used the tool to reduce queue waiting times at checkpoints in Terminal 1 at Naples airport. Their results provided key insights into

average queue times, along with minimum and maximum peaks, based on the number of available resources in the model. Similarly, Rusca et al. [13] applied the tool to simulate a hypothetical passenger terminal modeled after the main Romanian train station. The study showed how the simulation could be used to optimize the number of access gates, stairs, and waiting areas based on the collected data. In another case, Appelt et al. [14] used data from peak hours across different days to simulate passenger flow through kiosk, counter, and online check-in processes. Their ARENA-based model explored various scenarios, focusing on queue waiting times and total system times, allowing for an in-depth analysis of passenger throughput under different check-in modes. These studies demonstrate the Rockwell ARENA Simulation tool's effectiveness in modeling complex systems, offering actionable insights into resource optimization and system efficiency.

A map of Henri Coanda International Airport is presented in Figure 1.

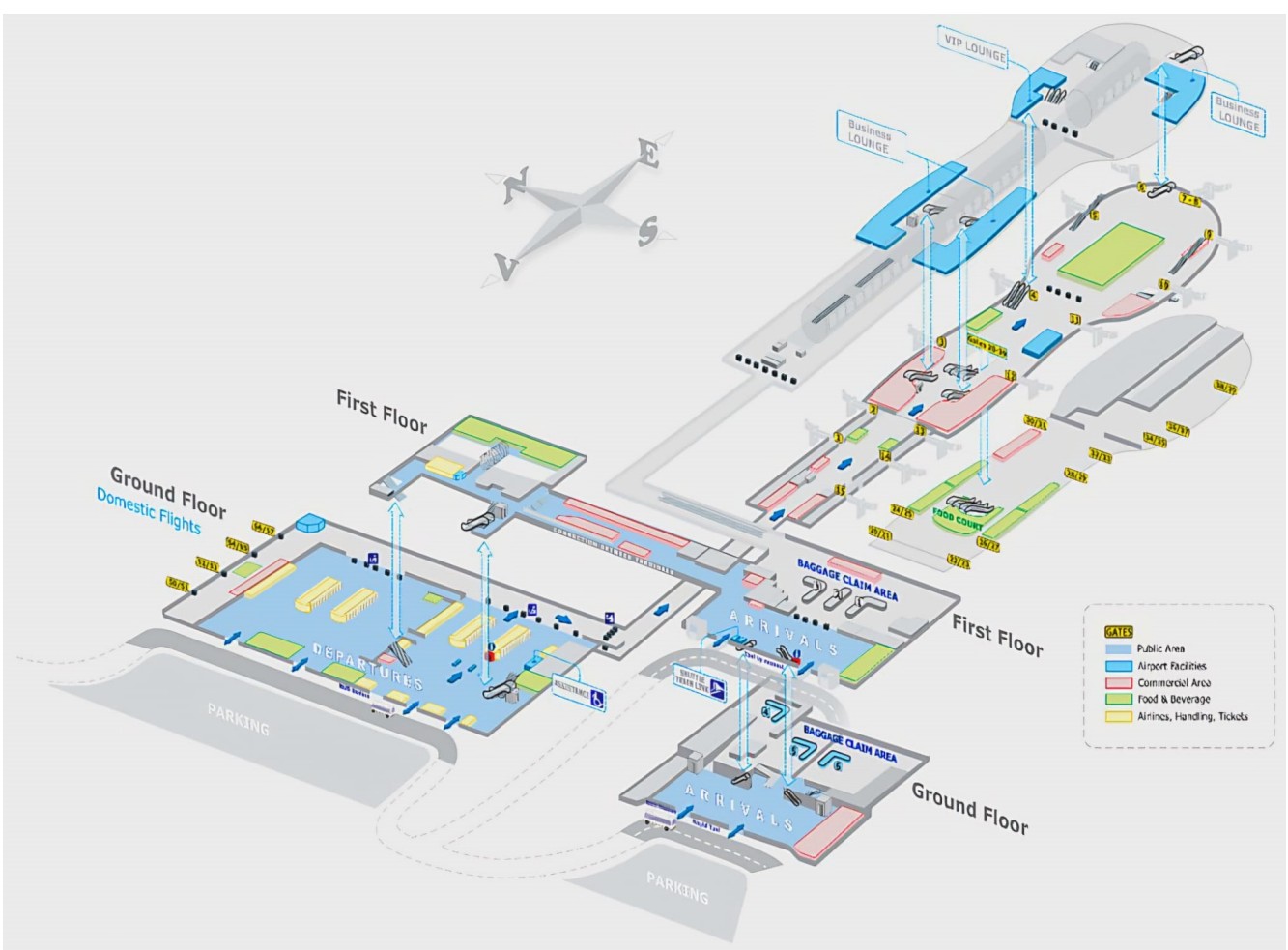

**Figure 1.** The map of Bucharest Henri Coanda International Airport (source: [15]).

Domestic flights from/to Bucharest Henri Coanda International Airport are presented in Figure 2.

The input data for the developed model are the domestic flights [17] and the number of passengers that depart from Bucharest Henri Coanda airport in the morning peak hour (7.00–9.00), presented in Table 1.

The model considers the characteristics of each passenger: flight number, departure time, with/without ticket, check-in online or in the airport, and with/without a checked-in bag.

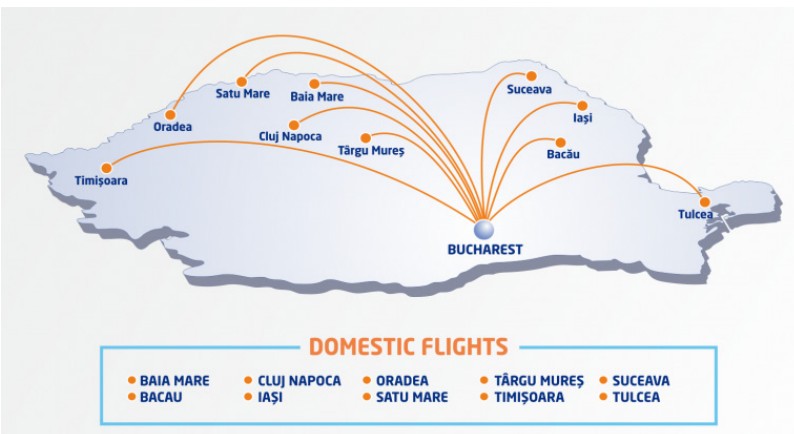

**Figure 2.** Domestic flights from/to Bucharest Henri Coanda International Airport (source: [16]).

The accuracy of the data used in modeling is crucial because it directly impacts the reliability of the results [18,19].

Figure 3 illustrates the sequence of actions and flows within the analyzed airport terminal, highlighting the decision points involved in the various processes accounted for by the simulation model. The diagram provides a detailed visualization of passenger movement through key areas such as the check-in, security check, and boarding gates. Each decision point represents critical junctures where choices are made, influencing the overall flow and efficiency of operations within the terminal. This model helps in understanding and optimizing the intricate dynamics of passenger processing and crowd management.

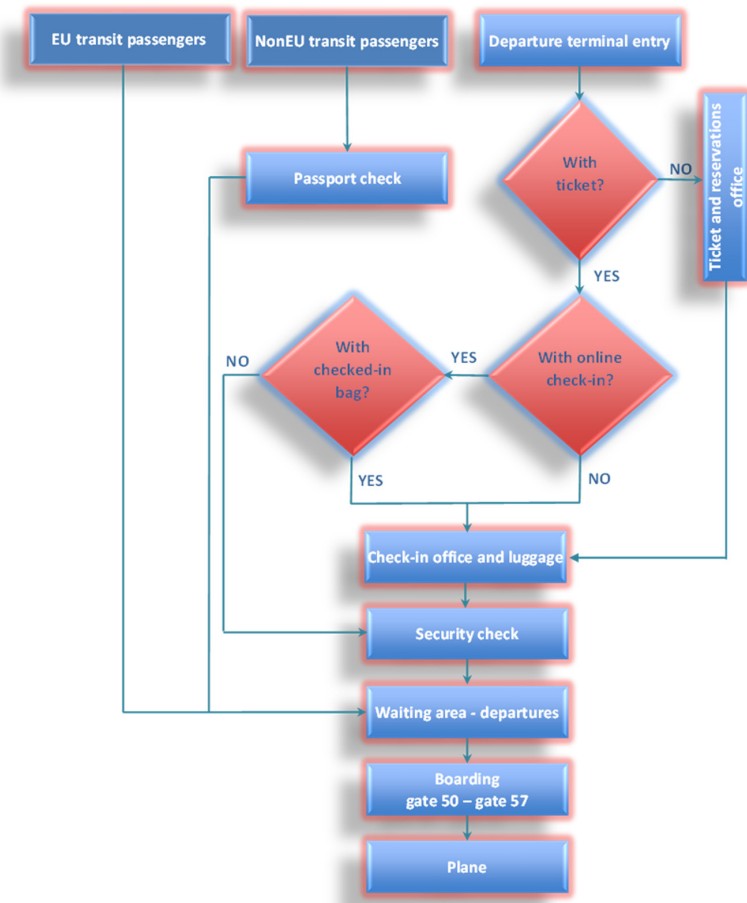

**Figure 3.** The diagram illustrating the departure process for domestic flight passengers (source: authors).

**Table 1.** The departures from Bucharest Henri Coanda International Airport (regular airline flights) (source: [20,21]).

| Airline | Destination | Departure Time | Aircraft | Number of Seats | Occupancy Rate | Transit Passenger Rate | Number of Passengers Originating from Bucharest |
|---|---|---|---|---|---|---|---|
| Group G1 | TIMISOARA (TSR) | 07:00 | Airbus 320-232 | 180 | | | |
| Group G2 | TIMISOARA (TSR) | 07:20 | Boeing 737-78J | 116 | | | |
| Group G2 | ORADEA (OMR) | 07:50 | ATR 72-600 | 74 | 85–100% | 5–10% | 90–95% |
| Group G2 | IASI (IAS) | 07:50 | Boeing 737-82R | 160 | | | |
| Group G2 | CLUJ NAPOCA (CLJ) | 08:10 | Airbus A318-111 | 113 | | | |
| Group G1 | CLUJ NAPOCA (CLJ) | 08:20 | Airbus 320-232 | 180 | | | |

### 4.2. The Simulation Model

The model of passenger flows within the airport terminal was implemented in ARENA 12 simulation software. The simulation program follows the diagram of departing passengers for domestic flights described in Section 4.1. In ARENA 12, the passengers are defined as entities, while the check-in and security desks are defined as resources. The passengers are passing through different modules of the simulation model, representing the defining/changing characteristics of the passengers, decision blocks, and states of the resources. The logic model is structured in three sub-models: passenger's arrivals (A), check-in (C), and security control (S) processes (Figure 4).

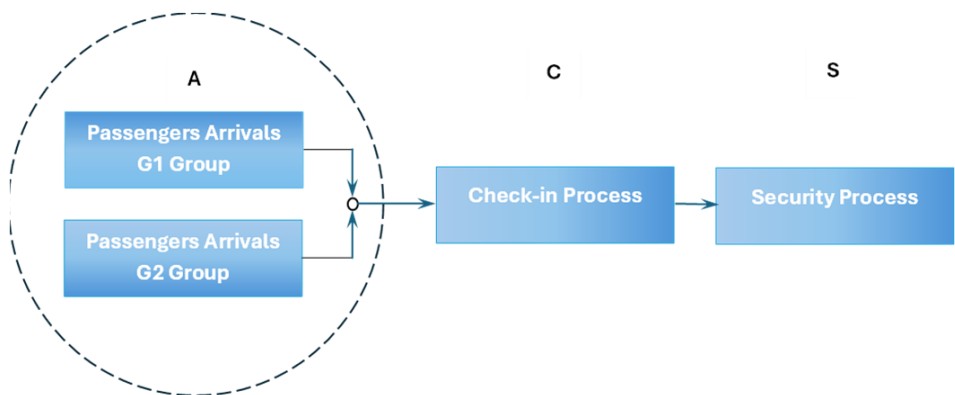

**Figure 4.** The structure of the logic model (source: authors).

In the first sub-model (Figure 5), passengers are generated based on a time distribution between their arrivals within the module Passenger Arrivals. The passengers start arriving at the airport 2.5 h before the departure of the airplane. The moment of a passenger's arrival at the airport is recorded to determine the time spent until the security check is finished.

The next step is to examine if the passengers bought their ticket online or whether they need to buy it from an office. In the second sub-model (Figure 6), they will follow the check-in procedure. If the passenger has already checked-in online and does not have luggage to check, then (s)he proceeds directly to the security control area. If the passenger has not checked-in online or (s)he has checked-in online, but needs to check the luggage, (s)he must wait for an available check-in desk. The bags are weighed, and a tag with their destination is applied to each bag.

In the simulation, the passenger waits in a queue until one check-in desk (a resource in ARENA) is available, seizes it, and then releases it.

In Table 2, the logical blocks are described.

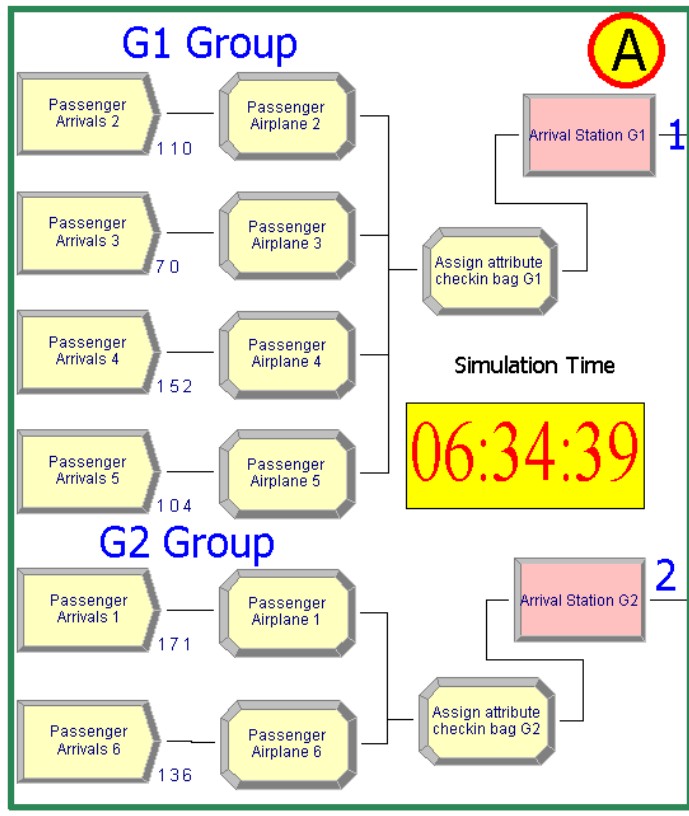

**Figure 5.** Passenger arrivals sub-model (source: authors).

**Table 2.** Description of simulation logical blocks.

| ARENA Module | Module Type | Role and Parameters |
|---|---|---|
| Passenger arrivals | Create | Each passenger is stochastically generated. The creation can be based on a schedule or on a time distribution between passengers. |
| Passenger airplane | Assign | The moment of a passenger's arrival at the airport is recorded to determine the time spent until the security check is finished. |
| Check-in bag G1/check-in bag G2 | Assign | Initially, all passengers have the attribute bag assigned with value 0. |
| Arrival station G1/arrival station G2 | Station | Defines a station corresponding to a logical location where processing occurs—route check-in G1/route check-in G2. |
| Online passenger ticket G1/online passenger ticket G2? | Decide | The module is used to model whether the passenger bought the ticket online or needs to buy it from an office. The decision is a "two ways by chance" type. |
| Buy ticket G1/G2 | Process | The process of buying a ticket is performed by one airport employee at the ticket desk. The process type is "seize–delay–release". |
| Online check-in G1?/online check-in G2? | Decide | The module is used to model whether the passenger made online check-in or needs to make it to a check-in desk. The decision is "two ways by chance" type. |
| With checked-in bag G1?<br>With checked-in bag G2? | Decide | The module is used to decide if the passenger with online check-in has or not bags to check-in. The decision is "two ways by chance" type. |
| Route check-in G1/route check-in G2 | Route | Transfers a passenger to Check in Station Module G1/Check in Station Module G2. A delay time to transfer to the next station is defined |
| Check-in G1 station/check-in G2 station | Station | Defines a station corresponding to a physical location where processing occurs—Check-in G1/Check-in G2 |
| Check-in G1/<br>Check-in G2 | Process | Each passenger occupies a check-in desk if is available. The process type is "seize–delay–release". The delay time considers the presence of bags for check-in. |
| Supplemental security desk? | Decide | The module is used to decide if it is necessary to open a new security desk. The decision is "two ways by condition", the condition being the queue length threshold. |
| New security desk opened | Assign | Increased the number of available security desk by one. |
| Security check | Process | Each passenger occupies a security desk if is available. The process type is "seize–delay–release". |
| Not necessarily supplemental security desk? | Decide | The module is used to decrease the number of security desks. The decision is a "two ways by condition" type, based on the queue security desk threshold. |
| Passengers to gate | Dispose | The passenger leaves the security checkpoint. This is the ending point for passengers in the simulation model. |

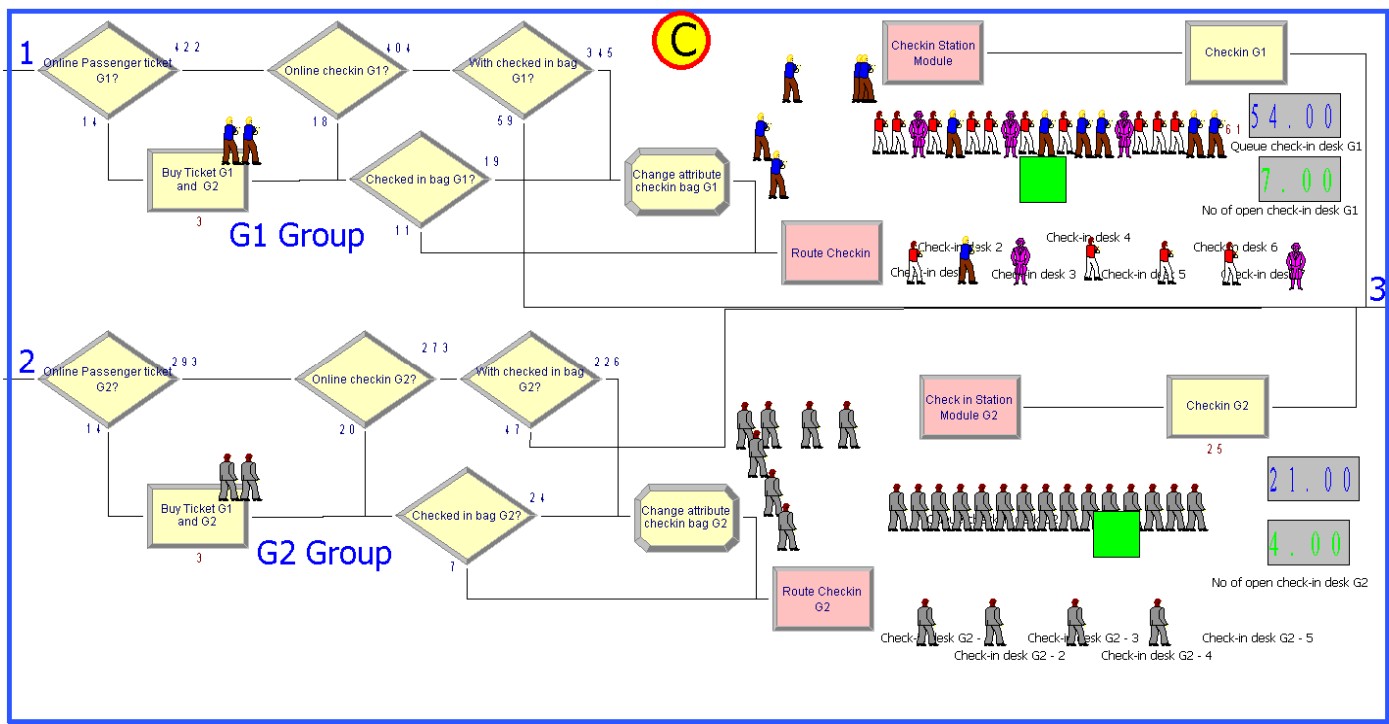

**Figure 6.** Check-in sub-model (source: authors).

After this process, all of the passengers proceed to the security control sub-model (Figure 7).

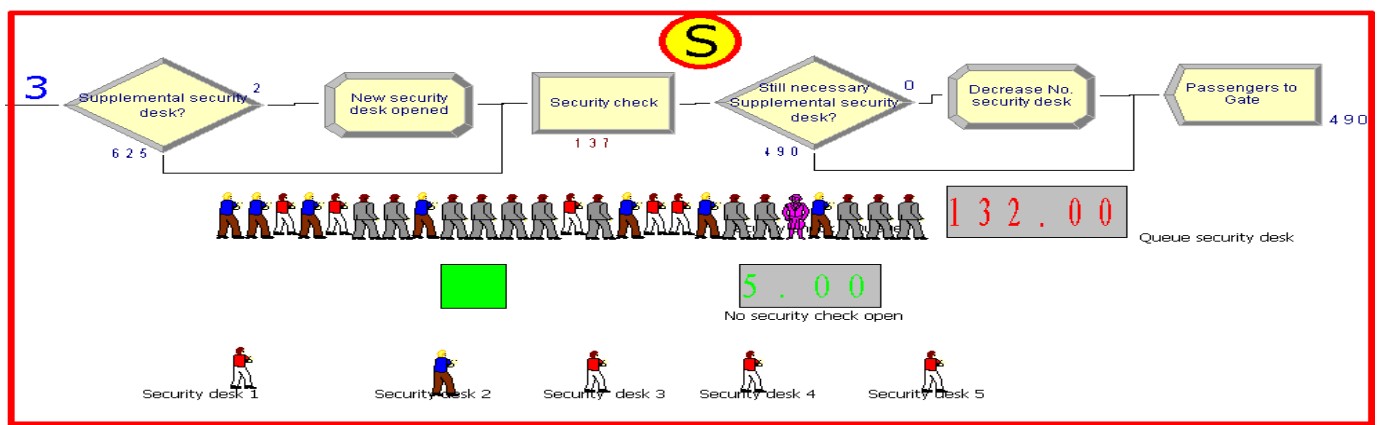

**Figure 7.** Security control sub-model (source: authors).

The accuracy of our simulation results is dependent on the quality of the input data. Due to limited publicly available information and strict privacy policies, we cannot access exact data from airports. As a result, we rely on distribution-based input data to approximate real-world conditions. Although this approach lacks the precision of actual measurements, it enables a generalizable and flexible simulation model. Furthermore, this methodology can be applied using precise data.

The research conducted by Lee et al. (2022), Brause et al. (2020), and Cui et al. (2018) on distributions of service time indicated that the triangular distribution is appropriate for service time for buying tickets and for security processes [22–24].

The triangular distribution density function is presented in Equations (1) and (2), with the following specific parameters: minimum (*a*), mode (*m*), and maximum value (*b*):

$$f(x) = 2\frac{x-a}{(m-a)(b-a)}, \; if \; a \leq x \leq m, \tag{1}$$

and

$$f(x) = 2\frac{b-x}{(b-m)(b-a)}, \; if \; m \leq x \leq b, \tag{2}$$

### 4.3. Experimental Results and Discussion

All airport processes (ticketing, check-in, security control) are characterized in addition to a specific service time by a queue waiting time. The waiting time in queues cannot be very high in order to ensure that all of the passengers that arrived in time to the airport will not miss their airplane flight.

The working scenarios implemented to optimize the number of check-in desks and security control inside the terminal Henri Coanda are:

- Scenario 1: Fixed number of security and check-in desks for G1 and G2 airline groups.
- Scenario 2: Variable number of security and check-in desks for G1 and G2 airline groups.

To investigate how the passenger flows within the airport terminal is functioning, a set of experimental input data were defined (Table 3) and used in ARENA simulation experiments.

**Table 3.** Simulation model input data.

| Model Entity/Resource | Data | Type | Values/Variation Range |
|---|---|---|---|
| Airplane A1–A6 | Number of seats | Constant | 180; 116; 74; 160; 113; 180 |
| | Occupancy rate | Constant | 85%, 90%; 95%; 100% |
| Passengers | First passenger generated at $t_0$ into the simulation run | Constant | t0 for A1 . . . 0; A2 . . . 20; A3, A4 . . .50; A5 . . .70; A6 . . .80 min |
| | Inter-arrival time | Triangular (min, mode, max) | A1, A6: TRIA (0.3, 0.33, 0.36) A2: TRIA (0.46, 0.51, 0.56) A3: TRIA (0.7, 0.81, 0.88) A4: TRIA (0.33, 0.37, 0.4) A5: TRIA (0.47, 0.53, 0.58) minutes |
| | Number of online ticket G1/G2 | Constant | 95% |
| | Number of online check-ins | Constant | 95% |
| | Number of check-in bags | Constant | 85% |
| | Travel time to check-in desk | Uniform (min, max) | Min = 2 min, max = 3 min |
| | Check-in process time | Normal (μ,σ) + constant | mean = 1 min; deviation = 0.2 min; time to weighed and tagged bag = 1 min |
| | Security process time | Triangular (min, mode, max) | TRIA (0.8, 1, 1.2) minutes |
| Ticket desk | Resource capacity | Constant | 1 desk |
| Check-in desk G1/G2 | Resource capacity | Variable | 3–7; 3–5 desks |
| Security desk | Resource capacity | Variable | 3–5 desks |

Due to the lack of historical data on passenger arrivals, which limited our ability to determine the best-fit distribution, we decided to use the triangular distribution for security processing time and the normal distribution for check-in processing time, accounting for the proportional impact of the number of bags to be checked. We assumed that all of the passengers would arrive within the interval $[t_0, t_0 + 90]$, where $t_0$ is two hours prior to the departure time of flights A1–A6, in order to minimize the risk of missing their flights.

For each scenario, 100 simulation replications were conducted. Each replication lasted 200 min, with the independent input data values selected within the range specified in Table 3.

The effectiveness of the passenger flows in the airport terminal is analyzed using three measures of performance (MOPs):

- The average passenger waiting time within check-in and security process is the mean time elapsed from the moment of the passenger joining the terminal resource queue to the moment of leaving that resource; it is a quality measure directly perceived by passengers, influencing their satisfaction, so smaller waiting times are desirable.
- The average passenger number queue length within check-in and security process.
- The average utilization rate of the check-in and security desk, a service parameter that illustrates the level of service of terminal resources.

### 4.3.1. Scenario 1—Fixed Number of Terminal Resources

In the first scenario, we ran simulations with different numbers of security and check-in desks for the G1 and G2 airline groups. These numbers are fixed for simulation time.

The evolution of the number of passengers in queues with six check-in desks for group G1, three check-in desks for group G2, and three security desks (6G1-3G2-3S) is depicted in Figure 8a. At the end of the simulation time, there are more than 200 passengers waiting for security processing, so more open security desks are necessary. As shown in Figure 8b, if two more security desks are open, then no more passengers will wait in the security queue, but there will still be some in the check-in queue in the G1 airline group (green line).

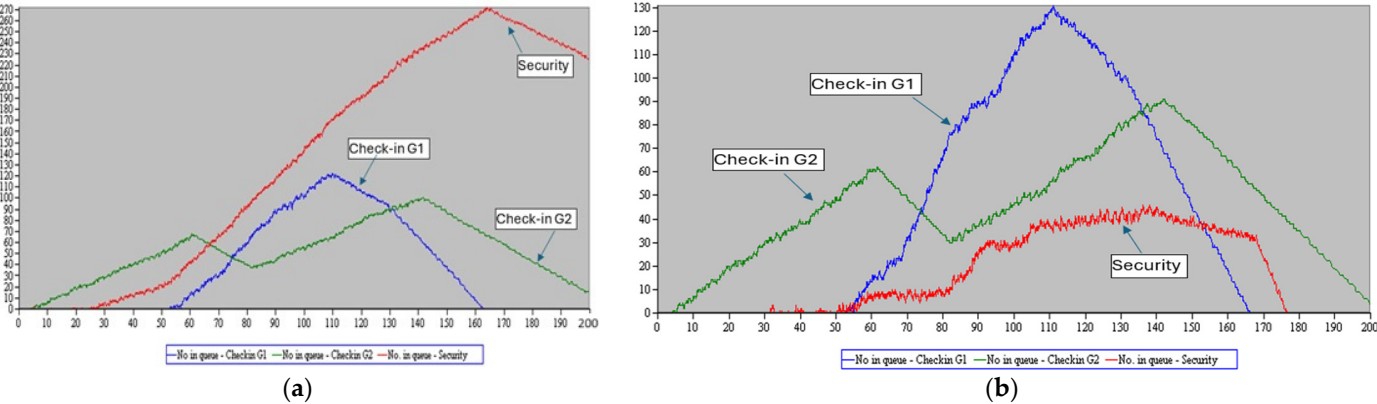

**Figure 8.** Queue length of check-in and security desk evolution for simulation 6G1-3G2-3S (**a**) and 6G1-3G2-5S (**b**).

Adding one more check-in desk for the G1 airline group leads to the dissipation of all queues, as shown in Figure 9.

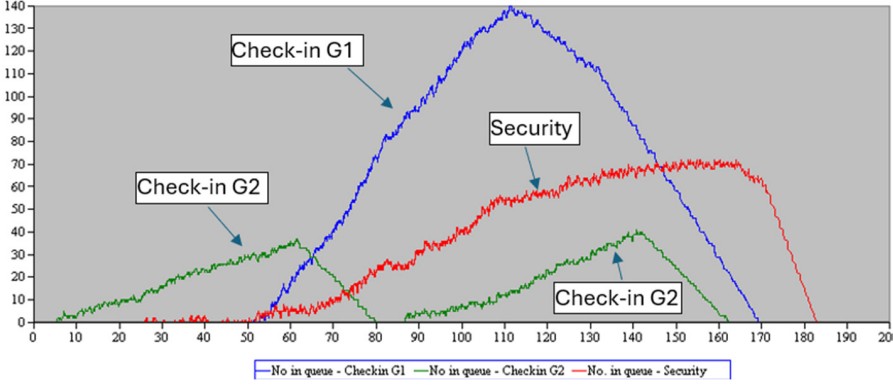

**Figure 9.** Queue length of check-in and security desk evolution for simulation 6G1-4G2-5S.

The solution 6G1-4G2-5S with six check-in desks for the G1 airline group, four check-in desks for the G2 airline group, and five security desks used the terminal's resources inefficiently, so we reduced by one the number of check-in desks for G2—Figure 10.

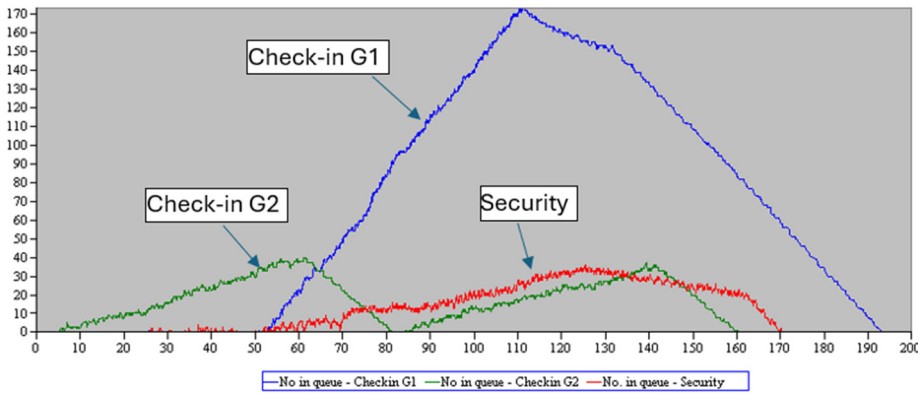

**Figure 10.** Queue length of check-in and security desk evolution for simulation 5G1-4G2-5S.

Figure 11 summarizes the results obtained in the different simulations in terms of minimizing the waiting times (a) or the number of passengers waiting (b) during the different processes within the airport terminal.

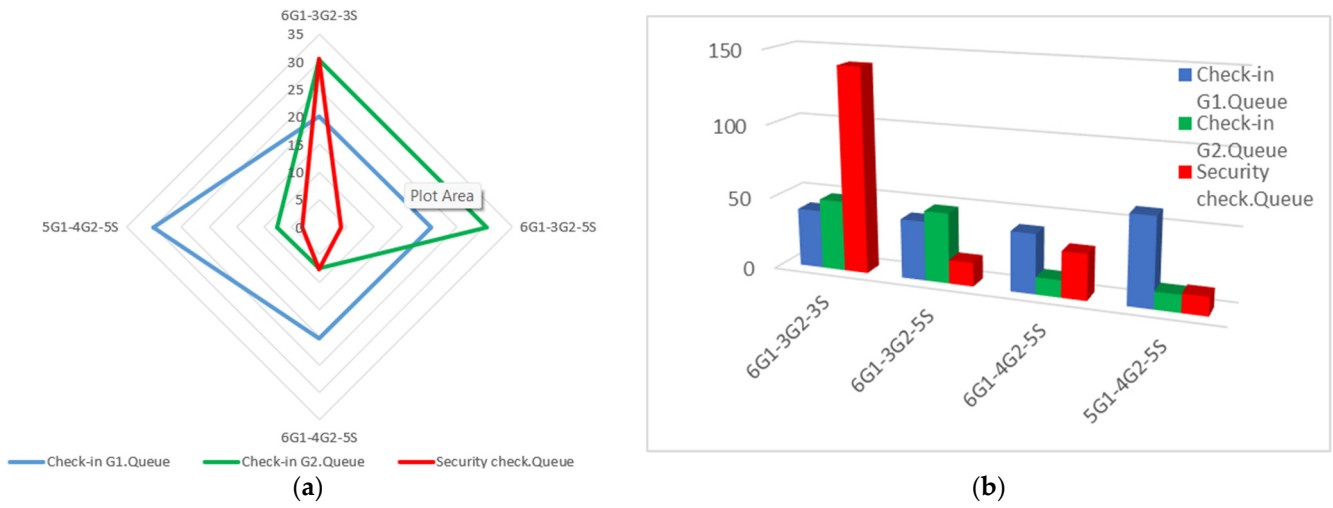

(**a**)                                                                     (**b**)

**Figure 11.** The average passenger waiting times (minutes) (**a**) and number waiting (**b**) within the airport terminal in Scenario 1.

The simulation 6G1-4G2-5S gives minimum check-in and security waiting times, but the utilization rate of the terminal resources (desks) is lower than the case with five security desks and between six and four check-in desks for the G1 and G2 airline groups (simulation 5G1-4G2-5S), as shown in Figure 12.

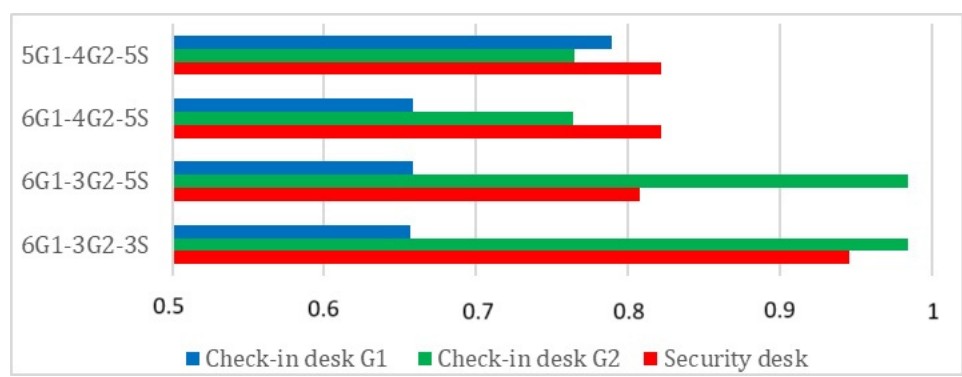

**Figure 12.** The utilization rate of the terminal resources in Scenario 1.

4.3.2. Scenario 2—Variable Number of Terminal Resources

In this scenario, we added or reduced the number of available check-in and security desks depending on whether they exceeded the threshold of the desk queue length, as mentioned in Table 3. As depicted in Figure 13(a1), the simulation 3-7G1 3-4G2 3-5S described as having from three to seven desks for check-in group airlines G1, from three to four desks for check-in group airlines G2, and from three to five desks for security procedures conducts to non-zero queue length at security desks at the end of the simulation run. To avoid this, we increased by one the number of available security desks and decreased by one the available check-in G1 desks, but, as shown in Figure 13(a2), this configuration 3-6G1 3-4G2 3-6S does not improve the final security queue, but just reduces the peak of this queue from an average of 71 passengers to 37. Also, we can observe in Figure 13(b1,b2) the variation in desk numbers for the above-mentioned simulations.

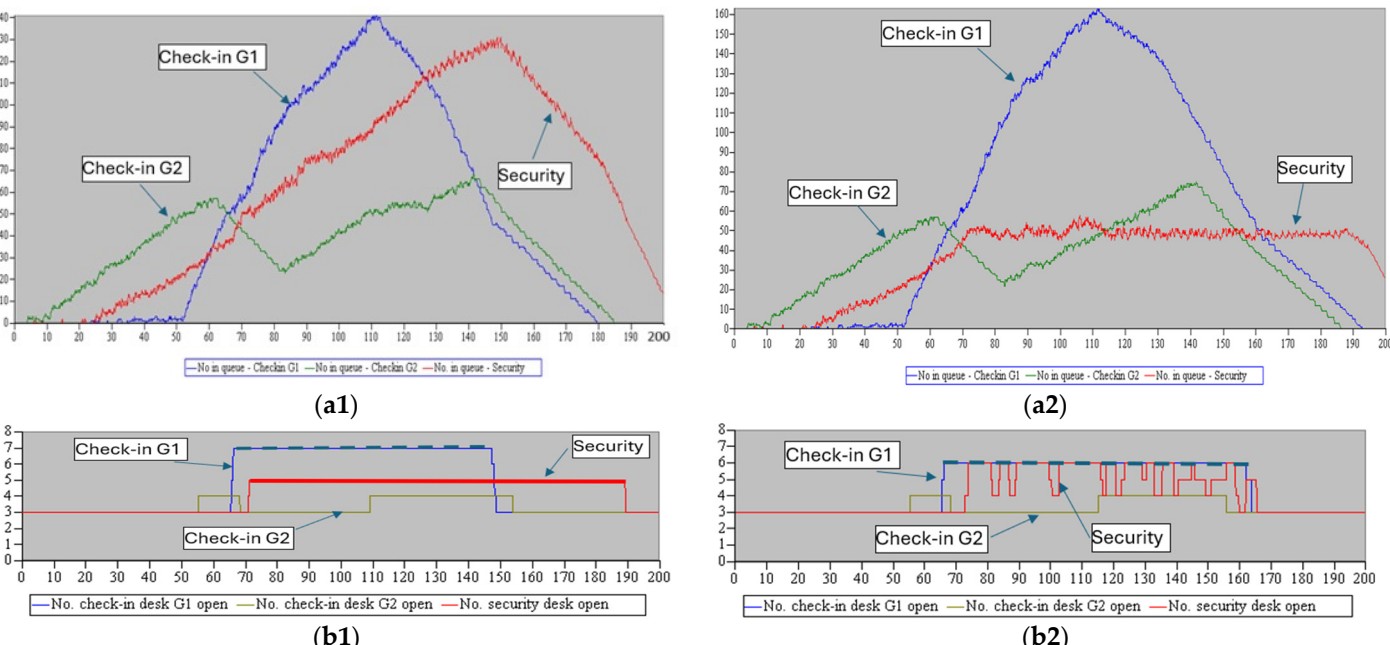

**Figure 13.** Evolution of queue length (**a**) and available number (**b**) of check-in and security desks for simulation 3-7G1 3-4G2 3-5S (**1**) and 3-6G1 3-4G2 3-6S (**2**).

If we maintained the maximum desks from the above simulations, we obtain the results in Figure 14 for simulation 3-7G1 3-4G2 3-6S. In this case, all queues are dissipated until the end of the simulation run.

We add one or two security desks in simulations 3-7G1 3-5G2 3-7S (3) and 3-7G1 3-5G2 3-8S (4) to find the optimal solution for the number desks required for the airport terminal (Figure 15).

As depicted in Figure 15(a3,a4), the queue length evolution of check-in and security desks are similar, but the rate utilization of security desks is better in configuration (3) according to Figure 15(b3,b4) and Figure 16.

The other two measures of performance, the average passenger waiting time and the average passenger number queue length within check-in and security process, are similar and lower than in the previously presented simulations (Figure 17).

Thereby, the airport desk configuration 3-7G1 3-5G2 3-7S (3), with three or up to seven check-in G1 desks, three or up to five check-in G2 desks, and three or up to seven security desks, gives the best results if we consider the three MOPs defined at the beginning of this section.

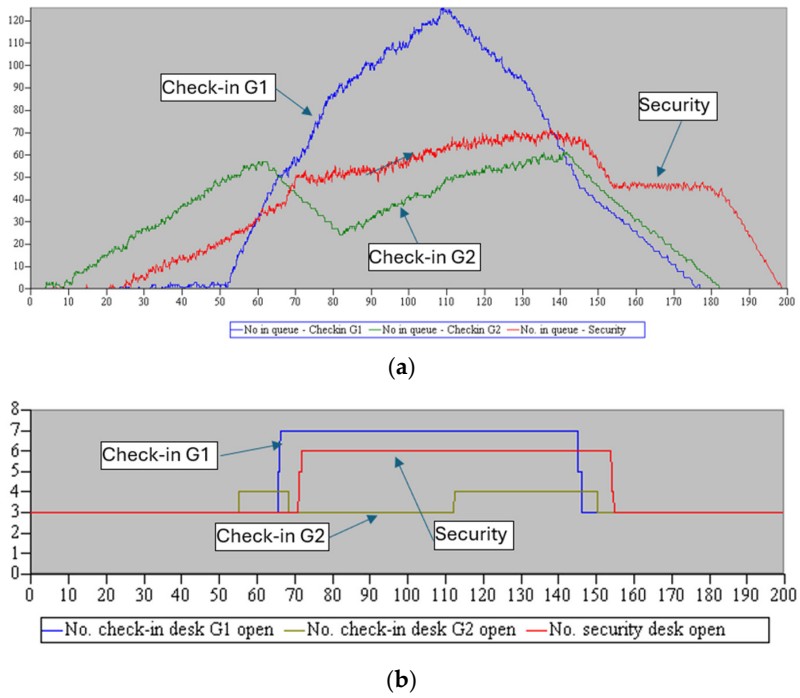

(**a**)

(**b**)

**Figure 14.** Evolution of queue length (**a**) and available number (**b**) of check-in and security desks for simulation 3-7G1 3-4G2 3-6S.

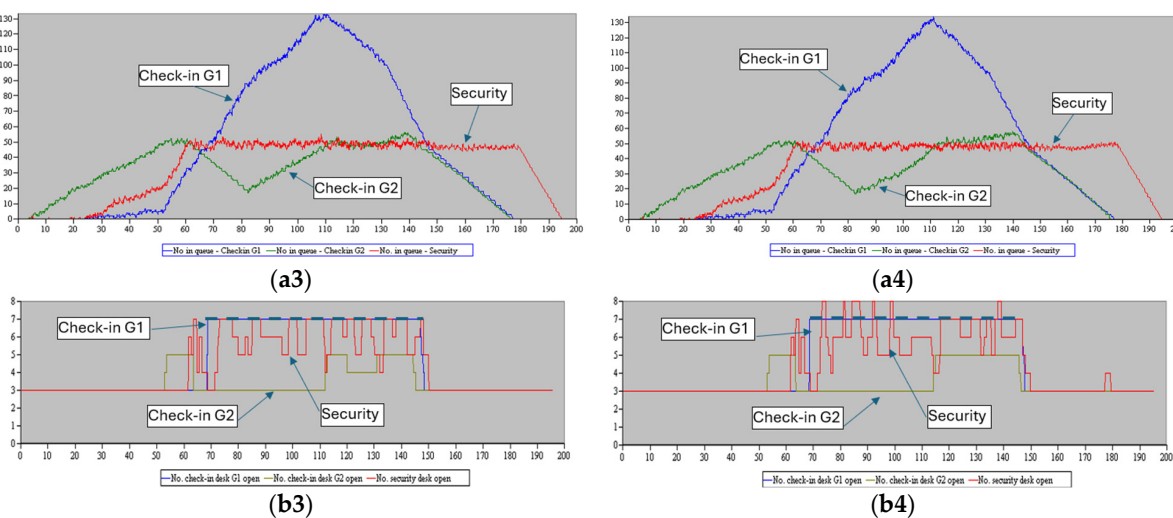

(**a3**)    (**a4**)

(**b3**)    (**b4**)

**Figure 15.** Evolution of queue length (**a**) and available number (**b**) of check-in and security desks for simulation 3-7G1 3-5G2 3-7S (**3**) and 3-7G1 3-5G2 3-8S (**4**).

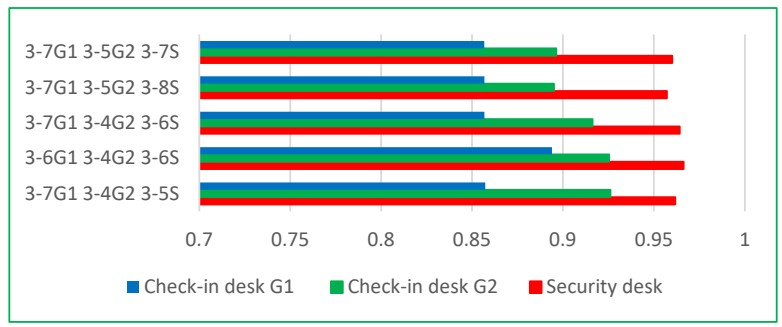

**Figure 16.** The utilization rate of the terminal resources in Scenario 2.

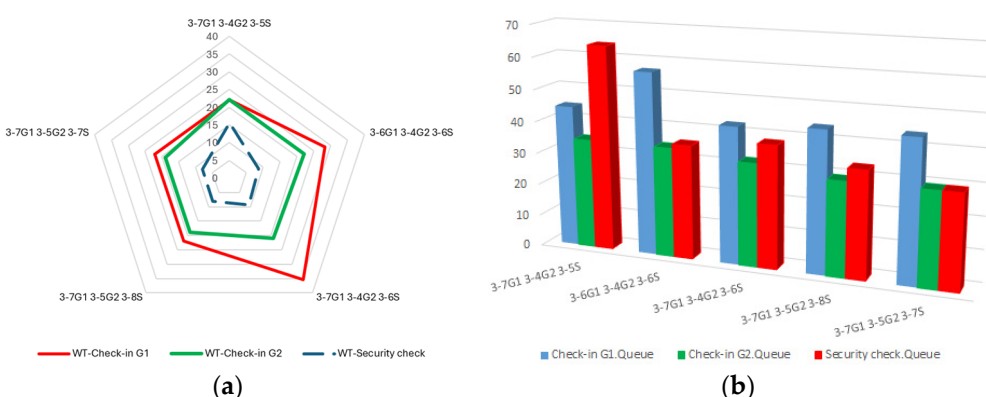

**Figure 17.** The average passenger waiting times (minutes) (**a**) and number waiting (**b**) within the airport terminal in scenario 2.

Figures 18 and 19 compare the average passenger waiting times and the average number of passengers waiting in Scenario 1 (hatched graphical representation) and Scenario 2 (plain graphical representation).

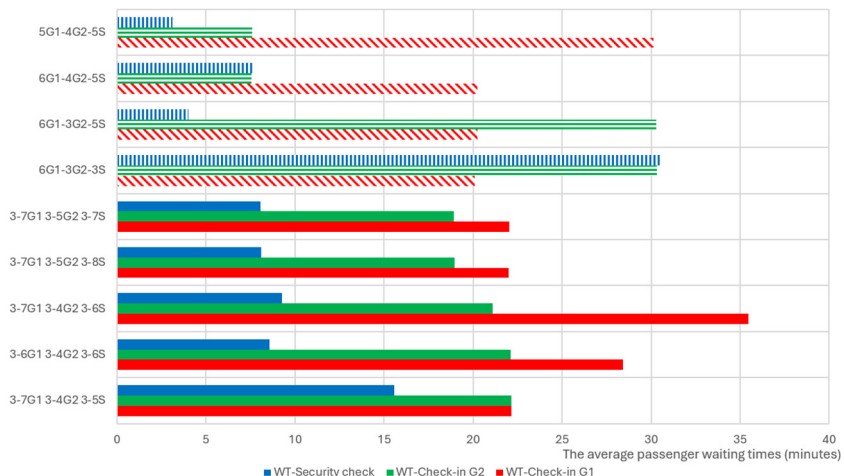

**Figure 18.** The average passenger waiting times in Scenario 1 and Scenario 2.

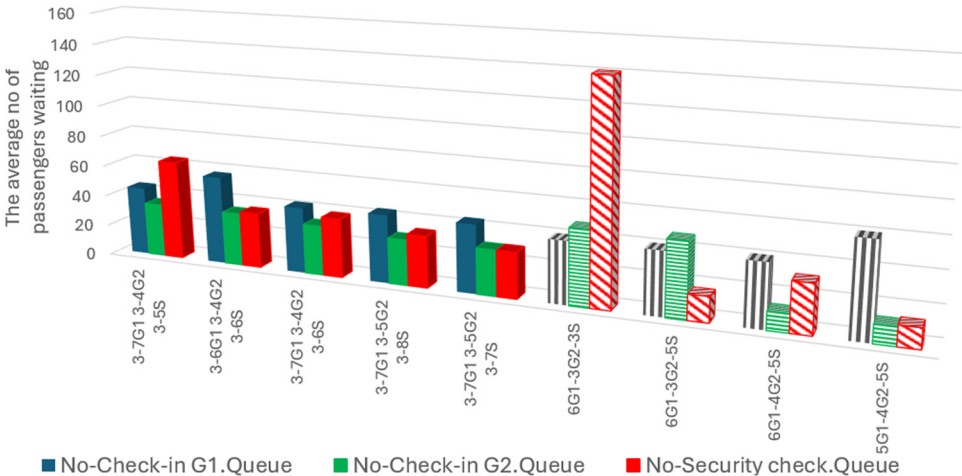

**Figure 19.** The average number of passengers waiting in Scenario 1 and Scenario 2.

Table 4 presents a comparison between the optimal desk configurations from Scenario 1 and Scenario 2, with the airplane occupancy rate set at 100%.

**Table 4.** Measures of performance comparison between the best configurations for Scenarios 1 and 2.

| Scenario/Simulation Measures of Performance | Scenario 1 5G1-4G2-5S | Scenario 2 3-7G1 3-5G2 3-7S |
|---|---|---|
| Average waiting time (minutes) | | |
| Check-in G1.Queue | 30.1 | 22.0 |
| Check-in G2.Queue | 7.6 | 18.9 |
| Security check.Queue | 3.1 | 8.1 |
| Average number waiting | | |
| Check-in G1.Queue | 60.3 | 44.0 |
| Check-in G2.Queue | 11.8 | 29.3 |
| Security check.Queue | 12.8 | 33.3 |
| Rate utilization | | |
| Check-in desk G1 | 0.79 | 0.86 |
| Check-in desk G2 | 0.76 | 0.90 |
| Security desk | 0.82 | 0.96 |

There is a relative increase in nearly all performance measures in Scenario 2 compared to Scenario 1.

## 5. Conclusions

This paper presents a simulation model designed to simulate passenger flow within an airport, addressing the challenges of managing high volumes of passengers and mitigating the associated risks.

A notable strength of the developed simulation model is its adaptability; it can be updated and modified to meet current requirements. This research involved studying passenger flows at Romania's main airport and identifying the key components of the simulation model, including operational processes, resource utilization, and service discipline at workstations. The variability in passenger volume was instrumental in establishing input data, enabling the accurate representation of interactions between flows and operations in the area studied. Two scenarios were examined: Scenario 1 maintains a constant number of security and check-in desks for the two airline groups, while Scenario 2 allows for a flexible allocation of these desks. The results indicate that Scenario 2 shows significant improvements across nearly all performance metrics (the average number of passengers waiting, the average passenger waiting times, and utilization rate) compared to Scenario 1.

The model demonstrates utility at the operational level by allowing for the terminal authorities to adjust the number of control teams in real-time based on incoming passenger flow. Its true value, however, lies at the strategic level, where it serves as a critical tool during the design phase of new airports or in projects aimed at enhancing airport security control areas. Utilizing a widely recognized software tool like ARENA 12 facilitates the standardization of the model, allowing for minimal calibration and application across any transportation terminal.

By incorporating dynamic elements and enabling scenario adjustments, this model contributes significantly to both academic research and practical applications. It provides a robust framework for exploring and implementing effective crowd management solutions in the complex environment of airport terminals. Thus, the flexible allocation strategy presented in Scenario 2 incorporates real-time adjustments that differ from static approaches often used in crowd management studies. This dynamic approach reduces passenger waiting times and queue lengths more effectively than static resource allocation, as evidenced by our performance measurements, thus supporting a more sustainable airport environment. Future research will focus on integrating additional process categories occurring within passenger terminals to identify critical and vulnerable areas more effectively.

This study faces limitations due to data accessibility and model complexity, making it challenging to implement a full optimization approach. The manual tuning of input variables was a practical solution given these constraints, allowing us to examine key scenarios without the additional complexity of formulating an optimization problem.

Future work could expand on this by developing a rigorous optimization framework to refine the insights provided by our initial simulations.

**Author Contributions:** Conceptualization, C.O. and M.R.; methodology, C.O. and E.R.; resources, I.C. and A.R.; writing—original draft preparation, C.O. and M.R.; writing—review and editing, A.I. and O.D.; project administration, C.O. All authors have read and agreed to the published version of the manuscript.

**Funding:** The research work contained in this paper was supported within the frame of the grant no. 113/04.12.2023 from the National Program for Research of the National Association of Technical Universities—GNAC ARUT 2023.

**Data Availability Statement:** Additional data can be obtained from the authors.

**Acknowledgments:** This work was supported by a grant from the National Program for Research of the National Association of Technical Universities—GNAC ARUT 2023.

**Conflicts of Interest:** The authors declare no conflicts of interest.

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
