# Peer review of "Analyzing Passenger Flows in an Airport Terminal: A Discrete Simulation Model"

_computation, doi:10.3390/computation12110223_

Round 1

Reviewer 1 Report

Comments and Suggestions for Authors

This study presents a simulation model aimed at evaluating crowd management strategies to enhance sustainability in airport operations. Focusing on passenger flow challenges within terminals and using ARENA simulation software, the study compares two scenarios at Henri Coanda International Airport: one with fixed resource allocation and the other with variable allocation. Results show that the scenario with variable allocation improves performance measures such as passenger waiting time, queue length, and resource utilization, contributing to sustainable airport management.

The article provides a well-structured and detailed analysis of airport crowd dynamics and techniques for efficiently handling passenger flows to improve sustainability. It represents a crucial step in advancing the field of simulation-driven and sustainable airport crowd management. However, there are major concerns surrounding presentation, writing, technical flow, and scientific methodology that need to be addressed before I can recommend its publication:

  • Page 2, Literature Background: In this section, the authors present previous relevant studies without any logical technical flow or progression. The authors should consider briefly describing the different directions in which previous studies have been conducted, their importance to the domain, and the corresponding results. 
  • Page 3, Section 4, Case Study: The authors mention using the Rockwell Arena Simulation tool to model the crowd dynamics in a terminal. Can they comment on the past uses of this tool in related studies and how the tool has been successful in capturing the dynamics of relevant real-life experimental data?
  • Page 4, line 1: The paragraph “Due to congestion… in airport terminals. Consequently, effective crowd management... seems more appropriate as a motivating point, and hence it is recommended to move from the case study section to either the introduction or literature survey section.

  • Page 4, paragraph 3: The authors refer to figure 4, which is two pages ahead and lies sequentially after another figure and table. The authors should maybe consider reordering figures for smoother technical flow.
  •  
  • Page 6, Figure 4: This figure presents the ARENA simulation model, and is difficult to grasp due to the excessive amount of details in the figure. To improve the concept clarity, the authors can present a bird's-eye view of the 3 modules for the two airline groups, and then show a zoomed-in, annotated image (with references to relevant places in the text) explaining the components of the 3 modules.
  • Page 8: Table 3, Simulation model input data: While describing the input data set, the authors assume a variety of distribution types (constant, triangular, and normal) for the model inputs. Can the authors add justification for the respective options and any relevant citations that support the distribution types used above? This is an important point in assessing the scientific rigor of this study, as the input distributions to the model can significantly impact the output trends and corresponding conclusions.
  • Page 11, Section 4.3.2: When studying the variable terminal resources case, the authors manually tune the input variables, i.e., the available security or checkin desks, to find the optimal scenario. However, a more rigorous approach to finding the optimal solution is to draft the case study as an optimization problem with the terminal resource inputs as input parameters and an appropriate objective function based on the outputs studied in the article. This reformulation leads to a more detailed, rigorous case study and may provide additional insights as to what terminal resource values emerge for optimal values of different objective functions.
  • Page 13, Paragraph 1: Instead of simply stating textually the difference in performance measures for the constant vs. variable terminal resources studies, the authors should consider overlaying the results in a single plot, to communicate the conclusions more effectively.
  •  
  • Page 13, Paragraph 2: The authors mention an airport desk configuration that gives the best results based on the performance measures considered in the study. Following up on the optimization reformulation mentioned above, the authors should consider running a parametric optimization, guided by algorithms, to provide additional support to this statement. Or else, the authors should clearly mention it as the best configuration in the exploratory variations considered in the article.
  • Either in this study or follow up studies, the authors should consider detailed ablation studies for the terminal resource parameters, i.e., in the variable resource case, given other parameters are constant, how does the performance metric vary with a change in the maximum checkin desk capacity?
  • Page 14, Section 5, Conclusions: The conclusion section is a bit disorganized and fails to summarize the key takeaways from the paper. In fact, the first few paragraphs fail to summarize the observations from the article, and the discussion only begins in the last few paragraphs. The authors should consider rewriting the conclusion to be concise and to the point.

Author Response

Reply to the Review Report:

  1. Page 2, Literature Background: In this section, the authors present previous relevant studies without any logical technical flow or progression. The authors should consider briefly describing the different directions in which previous studies have been conducted, their importance to the domain, and the corresponding results.

Reply:

Page 2-3 (lines 58-117): We changed literature background according to the reviewer’s suggestions.

  1. Page 3, Section 4, Case Study: The authors mention using the Rockwell Arena Simulation tool to model the crowd dynamics in a terminal. Can they comment on the past uses of this tool in related studies and how the tool has been successful in capturing the dynamics of relevant real-life experimental data?

Reply:

Page 3-4 (lines 149-163): We exemplified some successful use of the Rockwell Arena Simulation tool.

  1. Page 4, line 1: The paragraph “Due to congestion… in airport terminals. Consequently, effective crowd management... seems more appropriate as a motivating point, and hence it is recommended to move from the case study section to either the introduction or literature survey section.

Reply:

Page 1-2 (lines 46-49): We moved the paragraph to introduction. Thanks for the suggestion.

  1. Page 4, paragraph 3: The authors refer to figure 4, which is two pages ahead and lies sequentially after another figure and table. The authors should maybe consider reordering figures for smoother technical flow.

 Reply:

We removed the reference to that figure, as it was actually Figure 3, which had already been cited at line 178.

  1. Page 6, Figure 4: This figure presents the ARENA simulation model, and is difficult to grasp due to the excessive amount of details in the figure. To improve the concept clarity, the authors can present a bird's-eye view of the 3 modules for the two airline groups, and then show a zoomed-in, annotated image (with references to relevant places in the text) explaining the components of the 3 modules.

Reply:

Page 6-9: We replaced the original Figure 4 with four new figures, providing an overview (figure 4) and then detailing each module individually (figure 5-7).

  1. Page 8: Table 3, Simulation model input data: While describing the input data set, the authors assume a variety of distribution types (constant, triangular, and normal) for the model inputs. Can the authors add justification for the respective options and any relevant citations that support the distribution types used above? This is an important point in assessing the scientific rigor of this study, as the input distributions to the model can significantly impact the output trends and corresponding conclusions.

Reply:

Page 9-10, lines 226-234: We justified the use of distribution types (supported by some relevant citation) - The accuracy of our simulation results is dependent on the quality of the input data. Due to limited publicly available information and strict privacy policies, we cannot access exact data from airports. As a result, we relied on distribution-based input data to approximate real-world conditions. Although this approach lacks the precision of actual measurements, it enables a generalizable and flexible simulation model. Furthermore, this methodology can be applied using precise data.

  1. Page 11, Section 4.3.2: When studying the variable terminal resources case, the authors manually tune the input variables, i.e., the available security or checkin desks, to find the optimal scenario. However, a more rigorous approach to finding the optimal solution is to draft the case study as an optimization problem with the terminal resource inputs as input parameters and an appropriate objective function based on the outputs studied in the article. This reformulation leads to a more detailed, rigorous case study and may provide additional insights as to what terminal resource values emerge for optimal values of different objective functions.

Reply:

Our study faced limitations due to data accessibility and model complexity, making it challenging to implement a full optimization approach as suggested. The manual tuning of input variables was a practical solution given these constraints, allowing us to examine key scenarios without the additional complexity of formulating an optimization problem. While we recognize the value of such an approach for identifying optimal terminal resource allocations, it would require extensive data and a robust objective function formulation that goes beyond our current scope. Future work could certainly expand on this by developing a rigorous optimization framework to refine the insights provided by our initial simulations.

  1. Page 13, Paragraph 1: Instead of simply stating textually the difference in performance measures for the constant vs. variable terminal resources studies, the authors should consider overlaying the results in a single plot, to communicate the conclusions more effectively.

Reply:

Thank you for the suggestion. We have added Figures 18 and 19 to visually overlay the performance measures for the constant vs. variable terminal resources, as recommended, to enhance clarity and communication of our findings.

  1. Page 13, Paragraph 2: The authors mention an airport desk configuration that gives the best results based on the performance measures considered in the study. Following up on the optimization reformulation mentioned above, the authors should consider running a parametric optimization, guided by algorithms, to provide additional support to this statement. Or else, the authors should clearly mention it as the best configuration in the exploratory variations considered in the article. Either in this study or follow up studies, the authors should consider detailed ablation studies for the terminal resource parameters, i.e., in the variable resource case, given other parameters are constant, how does the performance metric vary with a change in the maximum checkin desk capacity?

Reply:

Thank you for the suggestion. As noted in our response to point 7, we plan to address this matter in future studies.

  1. Page 14, Section 5, Conclusions: The conclusion section is a bit disorganized and fails to summarize the key takeaways from the paper. In fact, the first few paragraphs fail to summarize the observations from the article, and the discussion only begins in the last few paragraphs. The authors should consider rewriting the conclusion to be concise and to the point.

Reply:

We have revised the conclusion section to make it more organized and concise, ensuring that it effectively summarizes the key takeaways from the paper. We hope you find these changes satisfactory.

Reviewer 2 Report

Comments and Suggestions for Authors

In this paper, the authors introduce a simulation model designed as a decision-making tool to assess and analyze various crowd management strategies with a focus on enhancing sustainability in airport operations. The results enrich the existing theoretical research models and have certain theoretical meaning. Therefore, I think the paper can be accepted. In addition, the following are the few comments, which may be included while revision.

1. Literature Background. The author 's review of the research status merely lists the work of these scholars without summarizing and analyzing them, pointing out their shortcomings, and then proposing the innovation of this paper.

2. The text of the legend in Figure 1 is too small, so it is recommended to enlarge it.

3. Figure 3 and Figure 4 should be supplemented with the necessary text description, not just graphics.

4. What is the source of the values and ranges of many variables used in Table 3 ? Is it consistent with the status quo ? Or is it consistent with existing research ?

5. The various lines in Figures 5-7 and 10-12 cannot be distinguished by black and white printing. It is recommended that the author use curves with small triangles, rectangles, etc. to draw them, so that they can be easily distinguished when black and white printing.

6. The text in Figures 8a and 14a should be placed horizontally.

7. The logical relationship between the analysis contents of each part needs to be further strengthened.

8. The list of references should be extended to include some recent papers as follow.

1) A multi-value cellular automata model for multi-lane traffic flow under lagrange coordinate.Computational and Mathematical Organization Theory. https://doi.org/10.1007/s10588-021-09345-w

2) An Evaluation of Passenger Satisfaction among Users of Huambo Airport in Angola.Urban Science, 2023, 7(2).

Author Response

Reply to the Review Report:

  1. Literature Background. The author 's review of the research status merely lists the work of these scholars without summarizing and analyzing them, pointing out their shortcomings, and then proposing the innovation of this paper.

Reply:

Thank you for the suggestion. We rewrite this chapter. We hope you find the changes satisfactory.

  1. The text of the legend in Figure 1 is too small, so it is recommended to enlarge it.

Reply:

Thank you for the suggestion. We have enlarged the Figure 1 for better readability.

  1. Figure 3 and Figure 4 should be supplemented with the necessary text description, not just graphics.

Reply:

Thank you for the suggestion. We detailed the figure 3 on page 5 and we replaced the original Figure 4 with four new figures, providing an overview (Figure 4) and then detailing each module individually (figure 5-7).

  1. What is the source of the values and ranges of many variables used in Table 3? Is it consistent with the status quo? Or is it consistent with existing research?

Reply:

Our study faced limitations due to lack of historical data on passenger arrivals to identify the best-fit distribution, thus we used the triangular distribution for security process time and Normal distribution for check-in process time (and proportional time with number of bags to check-in).

Page 10, lines 252-257: Due to the lack of historical data on passenger arrivals, which limited our ability to determine the best-fit distribution, we decided to use the triangular distribution for security processing time and the normal distribution for check-in processing time, accounting for the proportional impact of the number of bags to be checked. We assumed that all passengers would arrive within the interval [t0, t0+90], where t0 is two hours prior to the departure time of flights A1–A6, in order to minimize the risk of missing their flights.

  1. The various lines in Figures 5-7 and 10-12 cannot be distinguished by black and white printing. It is recommended that the author use curves with small triangles, rectangles, etc. to draw them, so that they can be easily distinguished when black and white printing.

Reply:

Thank you for your suggestion. We modified the figures by adding on each curve its signification in order to be distinguished in black and white printing.

  1. The text in Figures 8a and 14a should be placed horizontally.

Reply:

We modify the text orientation in Figures 11a and 17a.

  1. The logical relationship between the analysis contents of each part needs to be further strengthened.

Reply:

We inserted new details in the case study to strengthen the connection between the simulation model and theoretical concepts. We hope this improves the logical coherence of the manuscript.

  1. The list of references should be extended to include some recent papers as follow.

1) A multi-value cellular automata model for multi-lane traffic flow under Lagrange coordinate. Computational and Mathematical Organization Theory. https://doi.org/10.1007/s10588-021-09345-w

2) An Evaluation of Passenger Satisfaction among Users of Huambo Airport in Angola.Urban Science, 2023, 7(2).

 Reply:

Thank you for the suggested references. After careful review, we have incorporated the second paper into the Literature Background, as it closely aligns with our research focus. While we recognize the value of the first paper, its primary emphasis on traffic flow is outside the scope of our study, so we did not include it.

Reviewer 3 Report

Comments and Suggestions for Authors

In this paper the authors propose to use a commercial software to analyze the performance of an airport under the analysis of a set of inputs (occupancy of airplanes, arrivals of passengers, check-in process, etc.) with the aim of being used as a decision-making tool for example to improve the passenger experience or to achieve a better airport management. The proposed approach, involving pedestrian simulators and hypothetical scenarios to generate data, is commendable for its practical application. However, from my point of view, the manuscript predominantly is presented as a detailed report to solve a specific problem rather than advancing in a theoretical knowledge in the field, In addition, it is not clear the novelty of the proposed methodology. For this reason, my opinion is that the paper is not worth of publication. 

Furthermore, the paper can be improved by a more comprehensive integration of relevant literature. This would highlight the manuscript’s novelty and contribution to the field which is no clear to me after reading the paper. 

Next some specific comments are included:

·       The way in which the reference papers are cited in the literature review should be improved. The authors should follow the journal citation rules.

·       The aim of the paper should be stated early.

·       The authors should better justify the use of ARENA 12 simulation software.

·       It is not clear if the assumed parameters involved in the model comes from any demand study or are just an authors’ guess.

·       Some of the conclusions are general (true) comments which it is not clear that come from the analysis of the results provided in the paper.

·       The authors seem to state in the conclusions that the model implements the passenger behavior depending on their nationality. This is not clear to me.

·       The authors used a template for other journal.

 To sum up, I cannot recommend the paper for publication

Author Response

Reply to the Review Report:

  1. The way in which the reference papers are cited in the literature review should be improved. The authors should follow the journal citation rules.

Reply:

We rewrote the Literature Background chapter. We hope you find the changes satisfactory. Thank you for the suggestion.

  1. The aim of the paper should be stated early.

Reply:

Thank you for your suggestion. The aim of the paper is to develop and apply a simulation model for assessing and optimizing crowd management strategies in airport terminals, with a focus on enhancing operational efficiency, safety and sustainability under variable passenger flow conditions. We added this in the Introduction.

  1. The authors should better justify the use of ARENA 12 simulation software.

Reply:

Page 3-4 (lines 141-155): We justified the use of Rockwell Arena Simulation software.

  1. It is not clear if the assumed parameters involved in the model comes from any demand study or are just an authors’ guess.

Reply:

Our study faced limitations due to lack of historical data on passenger arrivals to identify the best-fit distribution, thus we used the triangular distribution for security process time and Normal distribution for check-in process time (and proportional time with number of bags to check-in).

Due to the lack of historical data on passenger arrivals, which limited our ability to determine the best-fit distribution, we decided to use the triangular distribution for security processing time and the normal distribution for check-in processing time, accounting for the proportional impact of the number of bags to be checked. We assumed that all passengers would arrive within the interval [t0, t0+90], where t0 is two hours prior to the departure time of flights A1–A6, in order to minimize the risk of missing their flights.

  1. Some of the conclusions are general (true) comments which it is not clear that come from the analysis of the results provided in the paper.

Reply:

  1. The authors seem to state in the conclusions that the model implements the passenger behavior depending on their nationality. This is not clear to me.

Reply:

We did not implement the model taking into consideration the passengers’ nationality. We have revised the conclusion section to make it more organized and concise, ensuring that it effectively summarizes the key takeaways from the paper. We hope you find these changes satisfactory.

  1. The authors used a template for other journal.

Reply:

We used the template from the Sustainability journal. The editors redirected us to the Computation journal. We will update the template if requested by the Computation journal’s editor.

We hope that the changes made to the paper, incorporating your suggestions and those of the other reviewers, have added significant value to the work. We kindly ask you to reconsider your recommendation regarding publication.

Round 2

Reviewer 1 Report

Comments and Suggestions for Authors

My comments have been satisfactorily addressed, and I can now recommend the publication of the article.

Author Response

Thank you!

Reviewer 3 Report

Comments and Suggestions for Authors

The authors have tried to improve the paper considering most of the comments raised by this reviewer. However, my first general comment was not even answered in the “response to the reviewer” report although I feel that it was improved in the manuscript. In any case, I think that my main concerns are not still well addressed mainly related to the contributions to the transportation field. 

I still feel that the paper is a good case study with good solutions, but more typical from a transportation consulting firm than for a research paper. Maybe I’m losing something, but I do not see a novel methodology that outperforms others proposed in research papers. In fact, the efforts made by the authors to justify the novelty and to review the literature, show that most of the cited papers come from conferences where the authors show the results of a case study. 

To sum up, if the authors explicitly show the novel contributions of the paper, clearly state how the proposed method outperforms others form the literature and that the approach can be of a general use in any airport, I may change my recommendation.

Author Response

Reply to the Review Report:

The authors have tried to improve the paper considering most of the comments raised by this reviewer. However, my first general comment was not even answered in the “response to the reviewer” report although I feel that it was improved in the manuscript. In any case, I think that my main concerns are not still well addressed mainly related to the contributions to the transportation field.

I still feel that the paper is a good case study with good solutions, but more typical from a transportation consulting firm than for a research paper. Maybe I’m losing something, but I do not see a novel methodology that outperforms others proposed in research papers. In fact, the efforts made by the authors to justify the novelty and to review the literature, show that most of the cited papers come from conferences where the authors show the results of a case study.

To sum up, if the authors explicitly show the novel contributions of the paper, clearly state how the proposed method outperforms others form the literature and that the approach can be of a general use in any airport, I may change my recommendation.

Reply:

Thank you for your insightful feedback and for acknowledging the improvements made in response to prior comments.

We understand that the primary concern lies in demonstrating the novel contributions of our work to the transportation field, as well as distinguishing it from case studies or consulting-based solutions. We have revised the manuscript to further highlight the unique contributions of our research and explain its generalizability and methodological advantages. Below, we address your points in detail:

1. Novel Contributions to the Field:

Our paper introduces a flexible and adaptive simulation model that differs from existing case study approaches in several keyways. While many studies focus on static or generic crowd management solutions, our model uniquely integrates dynamic resource allocation strategies tailored to varying passenger flow volumes. This adaptive component allows for the continuous optimization of resource allocation within airport terminals, which can be particularly valuable for decision-making in high-traffic periods or during operational disruptions. By focusing on adaptability and scalability, our model serves not only as a planning tool but also as a framework that can be applied across different terminal environments and passenger volumes, addressing both operational and strategic needs in airport management.

2. Comparative Advantages Over Existing Literature:

We recognize the importance of demonstrating how our approach outperforms or complements existing methodologies. The flexible allocation strategy presented in Scenario 2, incorporates real-time adjustments that differ from static approaches often used in crowd management studies. This dynamic approach reduces passenger waiting times and queue lengths more effectively than static resource allocation, as evidenced by our performance measures, thus supporting a more sustainable airport environment. Furthermore, our study’s use of ARENA 12 for simulation modeling allows for consistent standards across case studies, enabling straightforward adaptation and application to other airports or similar transportation hubs.

3. General Applicability Beyond the Case Study:

Although Henri Coanda International Airport served as our case study, the model was designed with broad applicability in mind. The key components—such as adaptive control of security and check-in desks, resource utilization analysis and real-time response to passenger flow variability—can be transferred to other airport terminals or transportation environments with similar crowd management needs.

4. Methodological Novelty and Literature Justification:

Our literature review was expanded to emphasize the gap that our model addresses in existing research. Many of the cited studies (both from journals - Urban Science, Journal of the Eastern Asia Society for Transportation Studies, Transportation Research Part C, Journal of Human Sciences, Sustainability, Transformation of Transportation, U.P.B. Scientific Bulletin, Journal of Air Transport Management, Urban Rail Transit, European Transport Research Review and from conferences) focus on case-specific solutions, whereas our model advances a method that combines simulation with adaptable parameters for both operational and strategic decision-making. This contribution, we believe, is a meaningful addition to transportation research.

5. Further Improvements and Future Research:

To address the limitations we encountered, we are currently exploring ways to automate input adjustments and incorporate a more rigorous optimization framework in future iterations of the model. By addressing these technical limitations, we hope to enhance the model’s applicability and reliability even further.

We have revised the manuscript to incorporate these points more explicitly and trust that this response will help clarify the novel contributions of our work and its relevance to both the academic and practical aspects of transportation research.

Thank you again for your valuable feedback.

Round 3

Reviewer 3 Report

Comments and Suggestions for Authors

Dear Authors:

Thank you for the, now yes, detailed responses to my comments. Although I still see more of a case study (and a good one) than a research paper in the proposed manuscript, I agree that it proposes some steps forward in the field. Therefore, I have changed my mind. Congratulations.